# Impact of Education on Metabolic Dysfunction-Associated Steatotic Liver Disease (MASLD): A Southern Italy Cohort-Based Study

**DOI:** 10.3390/jcm14061950

**Published:** 2025-03-13

**Authors:** Rossella Donghia, Caterina Bonfiglio, Gianluigi Giannelli, Rossella Tatoli

**Affiliations:** 1Data Science Unit, National Institute of Gastroenterology-IRCCS “Saverio de Bellis”, 70013 Castellana Grotte, Italy; catia.bonfiglio@irccsdebellis.it (C.B.); rossella.tatoli@irccsdebellis.it (R.T.); 2Scientific Direction, National Institute of Gastroenterology-IRCCS “Saverio de Bellis”, 70013 Castellana Grotte, Italy; gianluigi.giannelli@irccsdebellis.it

**Keywords:** MASLD, cohort study, education

## Abstract

**Background**: An association between education levels and liver disease has been confirmed, but not yet with metabolic dysfunction-associated steatotic liver disease (MASLD). The aim is to investigate the relationship between education and MASLD in two cohorts in southern Italy. **Methods**: The study cohort included 2909 participants assessed during the third recall of the MICOL study and the second of NUTRIHEP, subdivided into four groups based on education levels. **Results**: A strong protective association was found between MASLD and higher education levels. Participants had an OR = 0.50 (*p* < 0.001, 0.36 to 0.69 95% C.I.), OR = 0.29 (*p* < 0.001, 0.21 to 0.41), and OR = 0.24 (*p* < 0.001, 0.16 to 0.37 95% C.I.) for middle, high school, and graduate education, respectively. **Conclusions**: This study’s findings indicate that there is an association linking MASLD with education level, i.e., having a lower education level increases the risk of liver disease, and a proper policy to regulate education may also mitigate the ever-increasing problem of this disease.

## 1. Introduction

Liver disease affects an increasingly large proportion of the world population. It is currently the eleventh leading cause of death worldwide, accounting for 4% of all deaths [1]. In particular, the number of deaths among non-alcoholic fatty liver disease (NAFLD) patients doubled from 1990 to 2019 [2]. The burden of liver disease is expected to double over the period 2016–2030 due to the increasing incidence of major risk factors in an aging population [3,4,5].

Many people suffer from NAFLD in the world, and the significant epidemiological burden justifies the growing clinical and scientific interest in metabolic dysfunction-associated steatotic liver disease (MASLD) [6].

MASLD is the most recent definition of steatotic liver disease associated with metabolic syndrome [7]. Recent studies suggested an association between NAFLD and metabolic syndrome [8], defining NAFLD as the hepatic manifestation of metabolic syndrome [9]. The close association between NAFLD and metabolic syndrome rapidly showed the need to change the nomenclature [10]. The term MASLD was introduced in June 2023 in a multi-society Delphi consensus statement on a new fatty liver disease nomenclature, which led to the final withdrawal of the term NAFLD [11].

MASLD is characterized by fat accumulation in the liver, detected by imaging or biopsy, and is observed in individuals with little or no alcohol consumption, commonly affected by obesity, type 2 diabetes mellitus (T2DM), dyslipidemia, and/or hypertension [12].

A recent meta-analysis on the prevalence of MASLD in a general adult population, taking into account 92 studies, estimated a worldwide average prevalence of around 30%, with peaks around 44% in Latin America. In Western Europe, the prevalence is 25.1% (20.55–30.28%) [6].

The distribution of this liver condition in the world is heterogeneous and varies according to geographic region, ethnicity, genetic variants, and socioeconomic and lifestyle factors [3,13]. Historically, liver steatosis was less common in low-income countries [9]. The spread of metabolic diseases resulted in a significantly increased prevalence of liver steatosis in low- and middle-income countries as well [14].

There are currently extensive studies in the scientific literature evaluating a direct association between MASLD and socioeconomic factors. The aim of this study is to evaluate the impact of education levels on liver disease in a Southern Italian cohort.

## 2. Materials and Methods

### 2.1. Study Population

This observation study involved 2909 subjects living in two different small towns in the south of Italy (Figure 1).

This study involved two cohorts, MICOL and NUTRIHEP.

MICOL is a sub-sample from the Multicenter Italian study on Cholelithiasis (MICOL).

MICOL was started in 1985, enrolling subjects listed on the electoral register of Castellana Grotte in Southern Italy (*n* = 3500). The present study is based on data from MICOL IV (2013–2014), the third recall of MICOL subjects based on 1483 participants [15]. The methodological details of this population-based study have been described elsewhere [16,17]. All participants provided informed consent to take part before examination. The This study was approved in accordance with the ethical standards of the institutional research committee of the National Institute of Gastroenterology and Research Hospital “S. de Bellis” in Castellana Grotte, Italy (DDG-CE 782/2013. The date of approval was 15 April 2013, Prot. n.144/C.E. The study was conducted in accordance with the Helsinki Declaration of 1975 and adhered to the “Standards for Reporting Diagnostic Accuracy Studies” (STARD) guidelines (http://www.stard-statement.org/, accessed on 5 October 2019). This manuscript is organized according to the “Strengthening the Reporting of Observational Studies in Epidemiology-Nutritional Epidemiology” (STROBE-nut) guidelines https://www.strobenut.org/ (accessed on 5 October 2019).

For the second cohort, NUTRIHEP was used, involving 2195 subjects. This cohort was obtained in 2005–2006 from the medical records of general practitioners in the municipality of Putignano (>18 years) (Bari), a small city in Southern Italy about 20 km from the coast and 5 km from Castellana Grotte. Participants were first interviewed (*n* = 2550) in 2004–2005 by trained physicians to collect information on their socio-demographic and clinical characteristics and dietary habits [18]. From 2014 to 2016, all eligible subjects were invited to participate in the follow-up; adherence was 86.08% (*n* = 2195). All participants provided informed consent after receiving comprehensive information on the medical data under study. The study was approved by the Ethics Committee of the Ministry of Health (DDG-CE-502/2005; DDG-CE-792/2014, 20 May 2005 and 14 February 2014, respectively). The joint MICOL and NUTRIHEP cohorts were used, and finally 2909 subjects were analyzed (Figure 2).

This complete cohort was subdivided into healthy subjects and subjects with MASLD, as shown in Figure 2.

### 2.2. Lifestyle, Clinical, and Dietary Assessments

Lifestyle and anthropometric assessments were performed by clinicians during examination and single questions about lifestyle habits were administered. The level of education was expressed as years of schooling, categorized in four classes based on the Italian organization. Blood was collected from all subjects and stored in the biobank according to validated protocols and then processed by expert personnel. Liver disease was diagnosed by upper abdominal ultrasound and at least 1 out of 5 cardiometabolic parameters based on drug treatment or endocrinologist referral attesting the disease, so MASLD was built on diagnostic criteria of the Delphi consensus [11]. The Mediterranean Diet Score (rMED) was based on criteria in the literature [19].

### 2.3. Statistical Analysis

Subject characteristics are reported as mean and standard deviation (M ± SD) for continuous variables and as frequency and percentages (%) for categorical variables. To test the association between education level categories (primary school, middle school, high school, and graduates), the Chi-square test was used for categorical variables, while the Kruskal–Wallis equality rank test was used for continuous variables.

To evaluate the association of education levels with MASLD (Yes vs. No), logistic regression models, adjusted for age (categorized at 50 years) and gender (Model 1) or age, gender, smoking, job, marital status, income, and kcal intake (Model 2) were built. The estimators were reported as odds ratios (ORs) and relative 95% confidence intervals. Margins were calculated from predictions of a previously fitted model at fixed values of some covariates, and then a margins plot was built. Generalized Causal Mediation Analysis [20] was conducted to evaluate the direct, indirect, and total effects of MASLD on some covariates, using education level as the mediator.

To test the null hypothesis of no association, the two-tailed probability level was set at <0.05. The analyses were conducted with StataCorp. 2023. Stata Statistical Software: Release 18. College Station, TX, USA: StataCorp LLC., jamovi (version 2.3.28), and RStudio (“Kousa Dogwood” release).

## 3. Results

The socio-demographic characteristics of the cohorts are described in Table 1. Of the total cohort of 2909, 29.12% had completed primary school, 30.04% middle school, 30.66% high school, and 10.17% had a university degree.

The mean age in the total cohort was 59.69 ± 13.74 years, and the sub-cohort of younger participants (50.01 ± 13.34) in the category with the highest level of education showed a negative trend (*p* = 0.0001). Regarding gender, the distribution was almost homogeneous between the two sexes (50.43%), but with statistically significant differences between the groups (*p* < 0.001) and a prevalence of the female gender in the group with the highest level of education; 86.72% of the married or cohabiting participants had a middle school diploma, while widow/ers, due to their advanced age, had the lowest level of education (15.14%). Furthermore, the level of education is associated (*p* < 0.001) with jobs with greater responsibilities (23.65%) among subjects who have a degree, while 50.23% have a high school diploma. In comparison, retired individuals and housewives have a lower level of education (10.89% and 79.93%, respectively). The association with the level of education is also confirmed for income bracket (*p* < 0.001), where a higher level of education allows a higher income (>40,000 (EUR) (22.64%)). Regarding smoking habit, it is associated (*p* < 0.001) because more participants with a high level of education are non-smokers (90.51%), whereas regarding adherence to the Mediterranean diet (rMED), those with better adherence have a lower level of education (61.61% and 17.49%, respectively), mainly linked to local produce on the territory and the lower cost of food.

Blood and clinical parameters are described in Table 2.

All parameters, i.e., cardiac, anthropometric, and disease variables, showed a statistically significant variation among education categories (*p* = 0.0001). In fact, both systolic and diastolic pressure showed lower values in university graduates. Anthropometric parameters, such as waist and hip circumference and also body mass index (BMI), were worst in subjects with fewer years of education. Also, comorbidities, dyslipidemia, hypertension, diabetes, and MASLD prevalence were associated with a low education level (*p* < 0.001). The same trend was shown for glucose, insulin, homeostatic model assessment of insulin resistance (HOMA), aspartate amino transferase (GOT), hemoglobin A1C (HbA1c), alkaline phosphatase, ferritin, C-reactive protein (CRP), ceruloplasmin, and alpha-1 antitrypsin (α1AT), with statistically significant decreases (*p* < 0.05).

The association between education level and MASLD was examined in two different models adjusted for several covariates, as shown in Table 3.

In the first model, adjusted for age and gender, with primary school as the reference category, a strong negative association was shown with a high school education level (OR = 0.58, *p* < 0.001, 0.47 to 0.72 95% C.I.), and in graduates (OR = 0.52, *p* < 0.001, 0.38 to 0.70 95% C.I.). In Model 2, adding other covariates such as smoking, job, marital status, income, and kcal intake, the association became stronger, showing a protective role of education against MASLD. From middle school (OR = 0.50, *p* < 0.001, 0.36 to 0.69 95% C.I.), through high school (OR = 0.29, *p* < 0.001, 0.21 to 0.41 95% C.I.), to graduates (OR = 0.24, *p* < 0.001, 0.16 to 0.37 95% C.I.), the risk was halved. The margins plot is presented in Figure 3.

To provide insights into causal pathways, and break down effects for a more nuanced understanding, a mediation analysis was conducted with MASLD as the dependent variable, education as the mediator, and other socio-demographic parameters as covariates (Figure 4). Indirect components and direct and total effects are shown in Appendix A.

## 4. Discussion

When we examined the association between education level and the risk of developing MASLD in our two cohorts from Southern Italy, we found some interesting results, currently not yet investigated elsewhere in the literature.

Based on our results, subjects with middle school education had a 50% probability of developing MASLD, whereas this risk decreased in the model adjusted for some covariates to 29% with high school education. The higher education level, instead, reveals a protective role, with the odds of the disease reduced by about 76% in the ill group compared with healthy subjects, underlining the protective role of education in public health.

Chronic liver diseases are responsible for approximately two million deaths globally each year, one million of which are related to cirrhosis-related complications and one million to hepatitis and hepatocellular carcinoma [21,22]. As one of the most important indicators of socioeconomic status, educational attainment with a college degree has recently been demonstrated to be associated with a lower NAFLD risk [23,24]. Educational attainment plays a fundamental role in human health and has dual effects as a driver of opportunity but also as a reproducer of inequality. Promoting educational attainment should be incorporated into prevention policies to mitigate the risk of chronic liver diseases [22]. Understanding the relationship between education and health outcomes is key to reducing health disparities and improving population health. Thus far, only a few studies have focused on the association between education and chronic liver diseases [23,24]. The specific effects of education and income on long-term outcomes in fatty liver disease have not yet been sufficiently studied [25].

Recently, a cross-sectional analysis of the National Health and Nutrition Examination Survey (2017–2018) reported that US individuals with higher educational attainment were more likely to have a lower prevalence of NAFLD ascertained by vibration-controlled transient elastography measurements [23]. It is worth noting that in addition to the overall effect on health, education can also promote the acquisition of healthy lifestyle habits and adherence to healthy behavior, as well as a wide range of other capabilities and opportunities that can be applied to improve physical conditions [26]. Adults with higher educational attainment live healthier and longer lives compared to their less educated counterparts [27].

Understanding the role of the schooling process in health outcomes is relevant for health policies because it can show whether interventions should be aimed at increasing attainment, or whether it is more important to increase the quality of, change the content of, or otherwise improve the educational process at earlier stages for maximum health returns [27].

Another potential explanation is that education is a strong predictor of income and occupation [28]. People with higher education are more likely to be able to afford healthier lifestyles (e.g., consuming healthy food and exercising in gyms) and have better access to health care [29,30]. Additionally, less educated individuals could be more exposed to unhealthy food environments due to living in socioeconomically deprived areas with more access to fast food and takeaway outlets [31]. People with a higher education level tend to make healthier food choices, preferring fresh and minimally processed foods, i.e., to adhere more to the Mediterranean diet, because they are more likely to follow nutritional recommendations. In addition, they may have greater economic capacity to purchase quality foods, such as fish, nuts, and extra virgin olive oil, pillars of this diet. Social networks may also play a role, as people with similar socioeconomic positions often share similar lifestyle patterns [32].

Higher education levels, on the other hand, have been linked to a greater understanding of the importance of maintaining a healthy lifestyle and making informed choices [33]. Although the association between educational level and NAFLD has been investigated in a few studies, there are as yet no cohort studies based in Southern Italy.

## 5. Conclusions

The strength of this study is the large number of participants (*n* = 2909) from a cohort based on two municipalities in southeastern Italy.

Overall, this study is one of the first to investigate the relationship between MASLD and education levels, and shows a strong association between these two components. Therefore, this study has great societal significance. These findings highlight the importance of addressing socioeconomic factors in the prevention and management of MASLD, necessitating a comprehensive approach to promote equitable access to health care and education.

However, there are also some limitations. The main one is the analyzed cohort, limited to two small towns in Southern Italy. In the future, it could be useful to expand these observations to other regions of Italy, so that this study, if confirmed, can become representative of the Italian population, and possibly be able to compare this situation with other European countries. Furthermore, the addition of other social parameters, such as the area in which one lives and the number of members per family unit, could be useful to enrich this study.

## Figures and Tables

**Figure 1 jcm-14-01950-f001:**
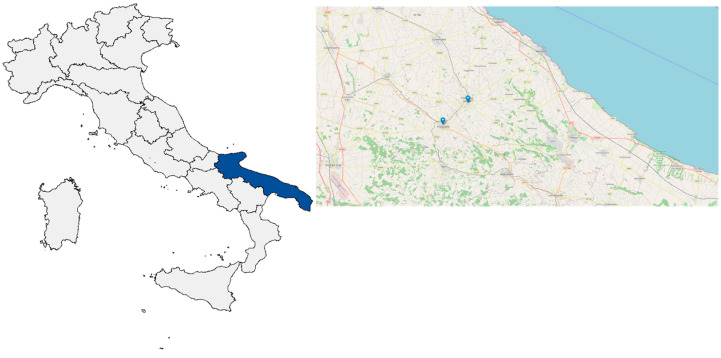
Geographical area of Southern Italy of the two towns involved.

**Figure 2 jcm-14-01950-f002:**
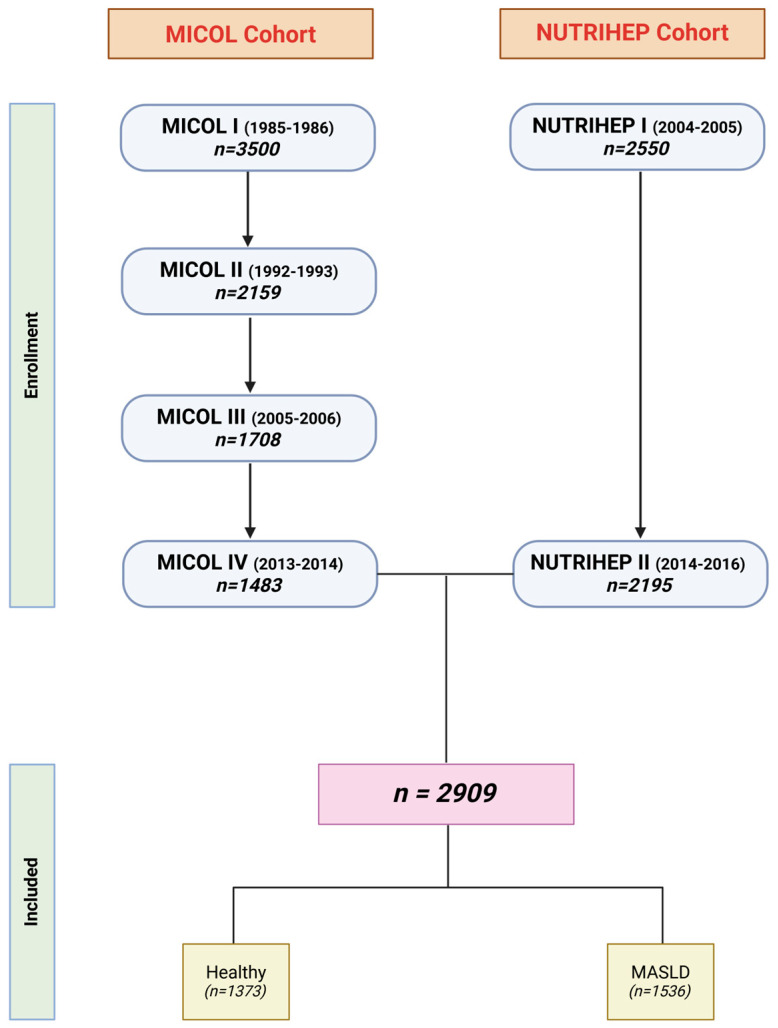
Flow chart of study population (created in https://BioRender.com).

**Figure 3 jcm-14-01950-f003:**
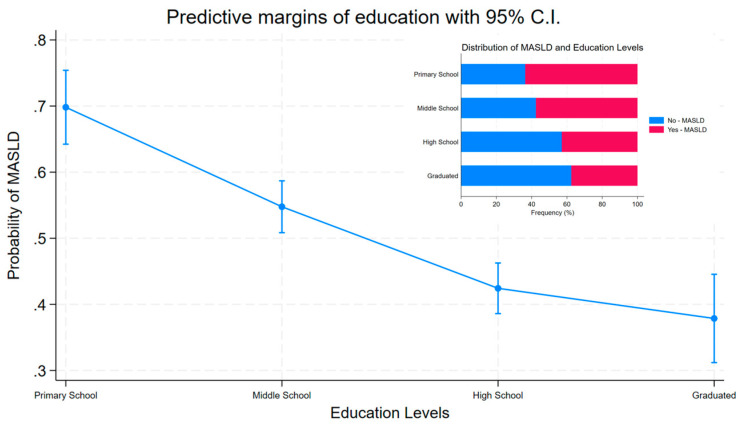
Predictive margins and distribution of MASLD by education level.

**Figure 4 jcm-14-01950-f004:**
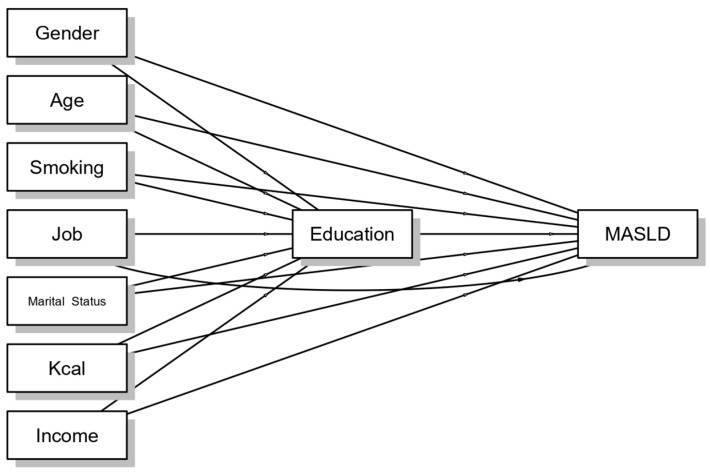
Mediation plot among covariates, education level, and MASLD.

**Table 1 jcm-14-01950-t001:** Socio-demographic characteristics of participants of the MICOL and NUTRIHEP studies.

Parameters *	Total Cohort (*n* = 2909)	Primary School (*n* = 847)	Middle School (*n* = 874)	High School (*n* = 892)	Graduate (*n* = 296)	*p* ^^^
Age (yrs)	59.69 ± 13.74	71.46 ± 8.09	58.79 ± 11.19	52.59 ± 12.57	50.01 ± 13.34	0.0001
Gender (%)						<0.001 ^¥^
Male	1467 (50.43)	381 (44.98)	477 (54.58)	467 (52.35)	142 (47.97)	
Female	1442 (49.57)	466 (55.02)	397 (45.42)	425 (47.65)	154 (52.03)	
Marital Status (%)						<0.001 ^¥^
Single	230 (8.52)	25 (3.10)	40 (4.92)	111 (13.74)	54 (19.85)	
Married or Cohabiting	2193 (81.25)	649 (80.52)	705 (86.72)	638 (78.96)	201 (73.90)	
Divorced or Separated	78 (2.89)	10 (1.24)	26 (3.20)	32 (3.96)	10 (3.68)	
Widow/er	198 (7.34)	122 (15.14)	42 (5.17)	27 (3.34)	7 (2.57)	
Job (%)						<0.001 ^¥^
Managers and Professionals	180 (6.20)	8 (0.94)	27 (3.10)	75 (8.45)	70 (23.65)	
Craft, Agricultural, and Sales Workers	904 (31.14)	48 (5.67)	259 (29.70)	446 (50.23)	151 (21.01)	
Elementary Occupations	324 (11.16)	53 (6.26)	181 (20.76)	89 (10.02)	1 (0.34)	
Housewife	219 (7.54)	55 (6.49)	95 (10.89)	61 (6.87)	8 (2.70)	
Retired Individuals	1168 (40.23)	677 (79.93)	269 (30.85)	168 (18.92)	54 (18.24)	
Unemployed	108 (3.72)	6 (0.71)	41 (4.70)	49 (5.52)	12 (4.05)	
Income (×year) (EUR) (%)						<0.001 ^¥^
<10,000	99 (3.40)	19 (2.24)	48 (5.49)	26 (2.91)	6 (2.03)	
10,000–20,000	623 (21.42)	180 (21.25)	249 (28.49)	169 (18.95)	25 (8.45)	
20,000–30,000	615 (21.14)	83 (9.80)	216 (24.71)	245 (27.47)	71 (23.99)	
30,000–40,000	262 (9.01)	9 (1.06)	53 (6.06)	135 (15.13)	65 (21.96)	
>40,000	195 (6.70)	3 (0.35)	25 (2.86)	100 (11.21)	67 (22.64)	
No Response	1115 (38.33)	553 (65.29)	283 (32.38)	217 (24.33)	62 (20.95)	
Smoking Habit (%)						<0.001 ^¥^
Never/Former	2524 (86.85)	771 (91.03)	754 (86.27)	732 (82.25)	267 (90.51)	
Current	382 (13.15)	76 (8.97)	120 (13.73)	158 (17.75)	28 (9.49)	
rMED (%)						0.03 ^¥^
Low	633 (26.07)	135 (20.90)	208 (28.42)	223 (28.41)	67 (25.28)	
Medium	1414 (58.24)	398 (61.61)	411 (56.15)	447 (56.94)	158 (59.62)	
High	381 (15.69)	113 (17.49)	113 (15.44)	115 (14.65)	40 (15.09)	

* Mean and standard deviation (M ± SD) for continuous variables, and frequency and percentage (%) for categorical variables. ^^^ Kruskal–Wallis rank test; ^¥^ Chi-square test. Abbreviations: rMED, Relative Mediterranean Diet.

**Table 2 jcm-14-01950-t002:** Blood and clinical parameters of participants of the MICOL and NUTRIHEP studies.

Parameters *	Total Cohort (*n* = 2909)	Primary School (*n* = 847)	Middle School (*n* = 874)	High School (*n* = 892)	Graduate (*n* = 296)	*p* ^^^
Systolic Blood Pressure (mmHg)	124.52 ± 15.94	132.05 ± 14.30	124.96 ± 15.36	120.07 ± 15.35	115.19 ± 14.05	0.0001
Diastolic Blood Pressure (mmHg)	78.33 ± 8.03	78.55 ± 7.88	79.27 ± 8.02	77.83 ± 8.06	76.38 ± 7.99	0.0001
Waist Circumference (cm)	91.66 ± 13.45	96.35 ± 12.59	92.98 ± 13.25	88.77 ± 13.38	86.66 ± 12.18	0.0001
Hip Circumference (cm)	102.27 ± 10.24	105.71 ± 10.54	103.05 ± 10.34	100.11 ± 9.56	99.25 ± 8.77	0.0001
BMI (kg/m^2^)	28.07 ± 5.18	29.73 ± 5.64	28.41 ± 4.90	26.82 ± 4.72	26.07 ± 4.25	0.0001
Dyslipidemia (Yes) (%)	576 (23.18)	256 (32.41)	171 (23.17)	117 (16.50)	32 (12.90)	<0.001 ^¥^
Hypertension (Yes) (%)	1049 (42.18)	509 (64.27)	297 (40.24)	201 (28.35)	42 (16.94)	<0.001 ^¥^
Diabetes (Yes) (%)	241 (9.70)	136 (17.17)	65 (8.83)	32 (4.51)	8 (3.23)	<0.001 ^¥^
MASLD (Yes) (%)	1536 (52.80)	539 (63.64)	503 (57.55)	383 (42.94)	111 (37.50)	<0.001 ^¥^
Blood						
Triglycerides (mg/dL)	103.01 ± 66.29	111.81 ± 67.58	101.52 ± 60.90	97.79 ± 67.79	98.79 ± 70.94	0.0001
Total Cholesterol (mg/dL)	191.76 ± 36.98	188.71 ± 38.41	192.16 ± 36.41	194.09 ± 36.23	192.28 ± 36.30	0.06
HDL (mg/dL)	50.16 ± 12.89	49.46 ± 12.76	49.77 ± 12.45	51.02 ± 13.42	50.74 ± 12.81	0.10
Glucose (mg/dL)	98.62 ± 22.17	105.01 ± 26.44	99.17 ± 21.03	94.28 ± 18.08	91.76 ± 17.91	0.0001
Insulin (mmol/L)	8.24 ± 6.28	8.61 ± 6.20	8.54 ± 6.81	7.94 ± 6.30	7.03 ± 4.19	0.0001
HOMA	2.12 ± 2.15	2.35 ± 2.30	2.21 ± 2.36	1.95 ± 1.96	1.67 ± 1.26	0.0001
GOT (U/L)	22.51 ± 14.82	22.65 ± 10.29	22.73 ± 14.74	22.49 ± 19.81	21.49 ± 5.61	0.86
SGPT (IU/L)	23.12 ± 16.68	21.97 ± 11.91	23.97 ± 19.59	23.61 ± 18.81	22.45 ± 11.11	0.003
HbA1c (%)	37.73 ± 7.17	40.17 ± 7.58	38.04 ± 7.45	36.88 ± 6.82	35.95 ± 5.74	0.0001
Total Bilirubin (mg/dL)	0.72 ± 0.37	0.70 ± 0.38	0.70 ± 0.35	0.72 ± 0.37	0.78 ± 10.91	0.01
GGT (U/L)	19.62 ± 17.62	20.26 ± 19.70	20.73 ± 20.88	18.56 ± 13.19	17.74 ± 10.91	0.07
Alkaline Phosphatase (U/L)	53.13 ± 16.07	57.12 ± 16.98	55.16 ± 15.94	50.46 ± 14.45	49.05 ± 16.86	0.0001
Albumin (U/L)	4.11 ± 0.27	4.04 ± 0.25	4.11 ± 0.26	4.14 ± 0.28	4.17 ± 0.26	0.0001
Iron (mg/dL)	89.93 ± 31.10	87.23 ± 27.99	89.32 ± 30.48	90.49 ± 32.36	93.64 ± 32.79	0.14
Ferritin (ng/mL)	100.80 ± 98.49	102.95 ± 91.77	105.69 ± 106.80	100.50 ± 99.50	85.75 ± 78.97	0.01
CRP (mg/dL)	0.26 ± 0.54	0.30 ± 0.49	0.28 ± 0.59	0.24 ± 0.52	0.21 ± 0.48	0.0001
Ceruloplasmin (mg/dL)	31.46 ± 7.44	32.06 ± 7.04	31.70 ± 7.04	31.45 ± 8.24	30.02 ± 6.28	0.0002
α1AT (mg/dL)	166.55 ± 36.34	172.32 ± 32.61	169.32 ± 35.66	164.31 ± 38.23	157.76 ± 35.38	0.0001

* Mean and standard deviation (M ± SD) for continuous variables and frequency and percentage (%) for categorical variables. ^^^ Kruskal–Wallis rank test. ^¥^ Chi-square test. Abbreviations: BMI, body mass index; MASLD, metabolic dysfunction-associated steatotic liver disease; HDL, High-Density Lipoprotein; HOMA, homeostatic model assessment of insulin resistance; GOT, aspartate amino transferase; SGPT, Serum Glutamic Pyruvic Transaminase; HbA1c, hemoglobin A1C; GGT, Gamma-Glutamyl Transferase; CRP, C-reactive protein; α1AT, Alpha-1 Antitrypsin.

**Table 3 jcm-14-01950-t003:** Logistic regression analysis of MASLD (Yes vs. No) in the MICOL and NUTRIHEP cohorts on education level, inserted as a single variable in the model ^^^.

	OR	se (OR)	*p*	95% C.I.
Model 1				
Education				
Primary School [Ref.]	--	--	--	--
Middle School	0.88	0.09	0.24	0.72 to 1.08
High School	0.58	0.06	<0.001	0.47 to 0.72
Graduate	0.52	0.08	<0.001	0.38 to 0.70
Model 2				
Education				
Primary School [Ref.]	--	--	--	--
Middle School	0.50	0.08	<0.001	0.36 to 0.69
High School	0.29	0.05	<0.001	0.21 to 0.41
Graduate	0.24	0.05	<0.001	0.16 to 0.37

Abbreviations: MASLD, metabolic dysfunction-associated steatotic liver disease; OR, odds ratio; se (OR), standard error of OR; 95% C.I., confidential interval at 95%. ^^^ Model 1 adjusted for gender and age (>50 yrs vs. <50 yrs); Model 2 adjusted for gender, age (>50 yrs vs. <50 yrs), smoking, job, marital status, income, and Kcal intake.

## Data Availability

The original contributions presented in this study are included in this article. Further inquiries can be directed to the corresponding author.

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
