# Peer review of "Impact of Education on Metabolic Dysfunction-Associated Steatotic Liver Disease (MASLD): A Southern Italy Cohort-Based Study"

_jcm, 2025, doi:10.3390/jcm14061950_

Round 1

Reviewer 1 Report

Comments and Suggestions for Authors

The paper by Donghi et al. deals with the increasing prevalent MASLD and with the known association between the level of education and the burden of the disease that is mostly caused by unhealthy life style leading to the metabolic syndrome. However, following points should be addressed:

-Use the term middle school instead of secondary as this may be confusing in certain countries: middle school, high school and graduate education

-lines 42-44, define the different risk factors as component of the metabolic syndrome and avoid the term “generally”

-line 55, I would suggest to replace “there are no study” with there are  no “extensive studies”

-lines 79-80, briefly mention whether the included population was based on hospital referred patients, general practitioner patients or general population

-line 92, don’t use the term “extracted” but obtained

-lines 141-143, please explain how liver disease was defined according to treatment (?) (a diagnosis cannot be properly inferred based on prescribed treatment) and what type of diagnosis was made by endocrinologist (diabetes, obesity?); moreover  check reference 20 that is not correct as it does not seem to be related to the Delphi consensus.

-In the table “parameters”, the level of income is not reported for 1967 (32%) patients, please explain why

-Lines 184-185, please explain: graduated individuals are significantly younger?

-Line 189, please define which specific age

-From the table parameters, remove the blood count ones as they are not relevant

-Line 252, MASLD is misspelled

-Line 254, nor fig. 4 but fig. 3

-Line 290, 24% risk of developing the condition does not mean reducing of 76% the risk of developing it, this statement is not correct

-Line 339, please mention the other limitations

-In the supplementary table, explain how MASLD is diagnosed (liver histology or?)

Author Response

The reviewer's response has been loaded as pdf document.

Reviewer 2 Report

Comments and Suggestions for Authors

The paper titled " Impact of Education on Metabolic Dysfunction-Associated Steatotic Liver Disease (MASLD): A Southern Italy Cohorts-Based Study" is interesting paper showing how education influence the impact on one of the most relevant health problems in noncomunicalbe desease. Paper is well written with good statistics and methods. There are some minor changes to be done:

Please check the apstract where you state  "This is the first study to demonstrate that educations levels are associated with MASLD" - it means that who has higher education has MASLD. Rewrite this sentence or explain it better. 

In Methods you mentioned two cohort  studies but you didn't explain how from these two cohort (enrollment) you have got the number 2909 patient (included). It is obvious that you have had some inclusion and exclusion criteria so you must explain them.

line 151 - MAFLD or MASLD?

In TABLE 1 THERE IS GENDER  IN BRACKET M than % - why it is important and if this is important where are women?

In Table 1 it is better to write "retired individuals" than "pensioners"  it is clear and professional.

Please try to avoid the linking words (overall, however, therefore,......). To much and in innappropriate way. 

It is interesting that you mentioned the Mediterrian diet score. If you find appropriate, please discuss also it in paper. 

Comments on the Quality of English Language

some phrases can be clearer.

Author Response

(The authors gave the same response as above.)
